# Maps of Constitutive-Heterochromatin Distribution for Four *Martes* Species (Mustelidae, Carnivora, Mammalia) Show the Formative Role of Macrosatellite Repeats in Interspecific Variation of Chromosome Structure

**DOI:** 10.3390/genes14020489

**Published:** 2023-02-14

**Authors:** Violetta R. Beklemisheva, Natalya A. Lemskaya, Dmitry Yu. Prokopov, Polina L. Perelman, Svetlana A. Romanenko, Anastasia A. Proskuryakova, Natalya A. Serdyukova, Yaroslav A. Utkin, Wenhui Nie, Malcolm A. Ferguson-Smith, Fentang Yang, Alexander S. Graphodatsky

**Affiliations:** 1Department of Diversity and Evolution of Genomes, Institute of Molecular and Cellular Biology, Siberian Branch of Russian Academy of Sciences, Novosibirsk 630090, Russia; 2State Key Laboratory of Genetic Resources and Evolution, Kunming Institute of Zoology, Chinese Academy of Sciences, Kunming 650223, China; 3Cambridge Resource Centre for Comparative Genomics, Department of Veterinary Medicine, University of Cambridge, Cambridge CB3 0ES, UK; 4School of Life Sciences and Medicine, Shandong University of Technology, Zibo 255049, China

**Keywords:** tandemly arranged sequences, macrosatellite DNA, telomere repeat, rDNA probe, fluorescence in situ hybridization, CDAG staining, sable, yellow-throated marten, pine marten, stone marten

## Abstract

Constitutive-heterochromatin placement in the genome affects chromosome structure by occupying centromeric areas and forming large blocks. To investigate the basis for heterochromatin variation in the genome, we chose a group of species with a conserved euchromatin part: the genus *Martes* [stone marten (*M. foina,* 2n = 38), sable (*M. zibellina*, 2n = 38*)*, pine marten (*M. martes,* 2n = 38), and yellow-throated marten (*M. flavigula*, 2n = 40)]. We mined the stone marten genome for the most abundant tandem repeats and selected the top 11 macrosatellite repetitive sequences. Fluorescent in situ hybridization revealed distributions of the tandemly repeated sequences (macrosatellites, telomeric repeats, and ribosomal DNA). We next characterized the AT/GC content of constitutive heterochromatin by CDAG (Chromomycin A3-DAPI-after G-banding). The euchromatin conservatism was shown by comparative chromosome painting with stone marten probes in newly built maps of the sable and pine marten. Thus, for the four *Martes* species, we mapped three different types of tandemly repeated sequences critical for chromosome structure. Most macrosatellites are shared by the four species with individual patterns of amplification. Some macrosatellites are specific to a species, autosomes, or the X chromosome. The variation of core macrosatellites and their prevalence in a genome are responsible for the species-specific variation of the heterochromatic blocks.

## 1. Introduction

In addition to unique sequences, eukaryotic genomes contain repetitive DNA that often forms heterochromatic blocks and remains densely packed throughout almost the entire cell cycle. Repetitive segments include both transposable elements and satellite DNA [1]. The sequences that make up the repeated-DNA fraction differ in their organization (tandem and dispersed), their prevalence in the genome (moderately or highly repeated), and their functional role. This part of the genome is now receiving more attention but remains largely enigmatic [2].

Tandem repeats are arranged head-to-tail in long arrays and constitute the satellite DNA fraction. Depending on the length of the monomer, satellite DNA is categorized into micro-, mini-, midi-, and macrosatellites. Of note, monomer sizes for each class vary among classifications by different authors [2,3,4,5,6,7]. These repeated sequences usually are harbored by centromeric, pericentromeric, and subtelomeric regions as well as by interstitial locations [8]. Satellite DNA takes part in different cellular processes, including chromosome segregation, chromatin conformation, and chromosome end protection [6,9]. Among the tandem repeats that affect the functioning of the genome are short telomeric sequences [10], moderately repetitive ribosomal genes [11,12], and highly repetitive noncoding microsatellite and satellite DNA [9]. Moreover, the largest of the tandem DNA repeats—macrosatellites—may represent both coding and noncoding sequences and have a structural and regulatory role in the organization of chromatin in the nucleus [5,13,14,15].

Cytogenetic data are suggestive of the involvement of constitutive heterochromatin in karyotype restructuring owing to colocalization with evolutionary breakpoint regions [16,17]. It has been shown that evolutionary breakpoint regions are often localized within segmental duplications and copy number variant regions and have a significantly higher density of all types of repeats [18,19,20]. Blocks of heterochromatin within a karyotype are hotspots of the chromosome rearrangements that may drive speciation events by contributing to reproductive isolation [21].

Most available genome assemblies are incomplete due to gaps at the locations of GC- and repeated-rich sequences [22]. Sequencing technologies and bioinformatic tools for the analysis of repeated sequences have already been developed [23,24,25,26], and recent technological advances showed an ability to assemble chromosomes from telomere to telomere [27,28]; however, the use of molecular cytogenetic approaches to study repeated DNA in mammals is still relevant.

In early work on rodent cytogenetics [29,30] and in studies on martens of the genus *Mustela* [31] and several canine species [32], it has been shown that the euchromatin part has the same size among the analyzed species and fluctuations in the size of the genomes are due to the addition of heterochromatic segments in the form of entirely heterochromatic arms.

The Mustelidae family unites animals that are diverse in morphology and behavior, belongs to the Caniformia suborder, and is one of the most specious in the Carnivora order [33]. Karyotypic studies of the genus *Martes* (Mustelidae) have revealed high conservatism of syntenic groups and similar G-banding patterns for many elements in chromosome sets among mustelids, raccoons, and cats [34,35]. Fluorescence in situ hybridization (FISH) has confirmed high conservatism of the euchromatin fraction among martens’ genomes in contrast to variation of the heterochromatin content. A comparison of human and domestic ferret karyotypes has uncovered a low number of conserved segments in total (32) [36,37,38,39,40] as compared to highly rearranged karyotypes (68) in the domestic dog [41,42].

Karyotypes of the species used in the present study do not have large heterochromatic segments and therefore have not attracted much attention in terms of research on repeated sequences. Nonetheless, a combination of comparative chromosome painting and different methods of differential staining makes it possible to obtain additional information on the ratio of euchromatin to heterochromatin for each element of the karyotype. In addition, based on the painting data, it is feasible to compare the accumulation of repeated sequences in homologous syntenic groups among related species.

Here we investigated constitutive heterochromatin in four *Martes* species by methods of bioinformatic analysis and molecular cytogenetics. Distributions of constitutive heterochromatin and of three types of tandemly repeated sequences (macrosatellites, ribosomal genes, and telomeric repeats) across genomes of the studied species are described too.

## 2. Materials and Methods 

### 2.1. Sampled Species and an Ethics Statement

Tissue samples from captive martens were used (Table 1). Ear biopsy was performed on a pine marten under inhalation anesthesia (isoflurane) during a veterinary examination in the Forest Fairy Tale Zoo of Barnaul (Russia). Fibroblast cell lines were established from a sable (from an ear and lung; Magistralny breeding fur farm, Altai Territory, Russia) and a yellow-throated marten (from an ear and lung; male Diksy, Rostislav Shilo Novosibirsk Zoo, Russia) using post mortem samples. The stone marten primary fibroblast cell line was obtained from the Center of Comparative Genomics of Cambridge University (UK). All the tissue samples were collected according to procedures approved by the Ethics Committee on Animal and Human Research at the Institute of Molecular and Cellular Biology, the Siberian Branch of the Russian Academy of Sciences (IMCB SB RAS; Novosibirsk, Russia; protocol No. 01/21 of 26 January 2021). Fibroblast cell lines from the domestic cat (*Felis catus*) and all animals studied here were deposited in the IMCB SB RAS cell bank (“The general collection of cell cultures,” 0310-2016-0002).

### 2.2. Cell Culture, Chromosome Preparation, and Differential Staining

The collection and transportation of the tissue samples, establishment of primary fibroblast cell lines, and chromosome preparation were performed as described before [44,45]. The stone marten and yellow-throated marten chromosome ID numbers correspond to nomenclatures by Nie et al. [39,43]. Chromosomes in the karyotypes of the sable and pine marten were arranged by length. Standard GTG-banding (G-bands by trypsin using Giemsa) [46] and the CDAG (Chromomycin A3-DAPI after G-banding) [47] method for revealing GC-enriched constitutive heterochromatin were employed. For each experiment (CDAG and detection of ribosomal DNA clusters, telomeric repeats, and macrosatellite repeated sequences [MSRs]), 8 to 18 GTG-stained metaphase plates were photographed, usually at least 12. 

### 2.3. Preparation and Characterization of Chromosome-Specific Painting Probes and Detection of Nucleolus Organizer Regions (NORs) and Telomeric Repeats

Whole-chromosome sorted painting probe libraries of the stone marten were used for FISH analyses of genomes of the sable and pine marten. The set of stone marten chromosome-specific painting probes has been described previously [39,43]. Amplification and labeling of a plasmid containing 18S, 5.8S, and 28S ribosomal DNA (rDNA) probes for detecting NORs and a probe containing telomeric sequences have been described in detail earlier [44].

### 2.4. Identification of Tandemly Arranged Repetitive DNA

Raw sequencing data of *M. foina* genomic DNA were downloaded from NCBI SRA under accession number SRX8108270. Filtering and trimming were performed using fastp 0.23.1 [48] with parameters “--detect_adapter_for_pe -5 -3 -r -l 50”. The trimmed filtered reads were analyzed in TAREAN 2.3.7 [25,48], which identified the most abundant tandem repeats. Consensus sequences were submitted to the NCBI GenBank database under accession numbers OQ261720–OQ261730. NCBI BLAST [49] was used to compare consensus tandemly arranged repeat sequences with available nucleotide sequences in the NCBI nr/nt database.

### 2.5. FISH, Image Acquisition, and Data Processing

The genomic DNA of *M. foina* was extracted by the standard phenol-chloroform method. Primers for PCR amplification and probes for labeling were designed using the primer3 software [50] (Appendix A). PCR amplification was performed as described in detail earlier [51]. Labeling was performed by PCR via the incorporation of biotin-dUTP and digoxigenin-dUTP (Sigma, Darmstadt, Germany). FISH was performed in accordance with previously published protocols [52].

Digital images of hybridization signals were captured using the VideoTest system (Zenit, St. Petersburg, Russia) and a Zeiss Axioscope 2 microscope (Zeiss, Oberkochen, Germany). The microscope was equipped with a charge-coupled device (CCD) camera (Jenoptik, Jena, Germany). Images of metaphase spreads were edited in Corel Paint Shop Pro Photo X2 (Corel, Ottawa, ON, Canada).

## 3. Results

### 3.1. Localization of MSRs in the Stone Marten Genome

First, bioinformatically identified MSRs found in the stone marten genome (Table 2) were mapped by FISH onto stone marten chromosomes to describe the distribution of the studied sequences in the genome of this species (Figure 1 and Figure 2). Each of the 11 MSRs hybridized with one to five sites in the stone marten karyotype. In autosomes, these sites are located in pericentromeric regions of 12 out of 18 pairs and in the short heterochromatic arm of chromosome 5. Four repeated sequences were found in 5p and various parts of the heterochromatic region on the Y chromosome. Notably, one of the repeats (S40) is X-specific, and the other, S46, is situated interstitially on the long arm of the X chromosome, in centromeres of several autosomes, and in the centromere of the X chromosome.

### 3.2. Homologous Elements in Karyotypes of the Stone Marten, Sable, Pine Marten, and Yellow-Throated Marten; Localization of Telomeric Sequences and Ribosomal Genes in Martes Species

To compare macrosatellite markers on homologous syntenic groups among the four *Martes* species, we relied on cross-species comparative painting data with whole-chromosome sorting probes from the stone marten. A genome-wide homology map between the stone marten and yellow-throated marten was created earlier [39]. These data indicated high chromosome conservatism and only one interchromosomal rearrangement—Robertsonian fission of MFO 7—in the karyotype of the yellow-throated marten. By means of the stone marten chromosome probes for the comparative painting of sable and pine marten chromosomes, we confirmed the earlier evidence of high chromosomal conservatism and one-to-one correspondence to stone marten chromosomes on the basis of comparative G-banding [34,35].

Here genome-wide homology maps of the stone marten and of the three other analyzed species were supplemented with information on the localization of ribosomal genes, telomere repeats, and MSRs. Clusters of rDNA are located on one chromosome pair in all the species under study: on autosome 15 of the stone marten and on homologous elements in genomes of the three other *Martes* species. Heteromorphism of the size of ribosomal gene clusters was found among the analyzed animals (Figure 3 and Figure 7).

Telomeric repeats were revealed at a terminal position only, with minor variations in cluster size (Figure 3).

### 3.3. Localization of MSRs in Genomes of the Pine Marten, Sable, and Yellow-Throated Marten

The MSRs utilized in this work may be subdivided into several groups (Table 3): 1. Stone marten–specific (S32 and S41), which were not found in the three other species. The remaining nine macrosatellites are present in the genomes of all four studied representatives of *Martes*. 2. Autosome-specific (S17, S18, and S26). 3. X chromosome-specific (S40). 4. Diverse: MSRs that were detected in centromeric regions of several autosomes, short arms of some autosomes, and one of the sex chromosomes (S22, S30, S35, S46, and S48). These species were found to differ in the location, fluorescent-signal intensity, number, and size of clusters of repeated sequences; in fact, each species has its own unique chromosomal “portrait” due to the diversity of the heterochromatin component (Figure 4, Figure 5 and Figure 6). We did not observe the heteromorphism of macrosatellite signals within each of the studied animals.

The number of identified localization sites varied among different MSR probes and among the four studied species. The sable, stone marten, and pine marten are comparable in terms of the number of heterochromatic segments. The yellow-throated marten genome has many more localization sites of tandem macrosatellite repeats, although the fluorescent signals and size of the identified clusters are smaller.

Letters “L” and “H” in repeat names indicate the prediction of TAREAN on the probability (L: low or H: high) of a tandem arrangement in the genome (Table 2). Our localization data suggest that all the analyzed sequences are arranged tandemly in the genome.

Because there is evidence of the conservatism of some MSRs among mammals [13], we tried to localize the X chromosome-specific satellites (S40H and S48H) and a pair of autosome-specific satellites (S22H and S26L) on the chromosomes of the domestic cat: the Carnivora species with a conserved genome. No signals from MFO macrosatellites were found on domestic-cat chromosomes in experiments on distant FISH, possibly indicating either the recent origin of these macrosatellites after radiation of Feliformia and Caniformia or a high degree of repeat divergence. The negative FISH results on the domestic cat are presented in Supplementary information (Appendix A).

### 3.4. A Comparison of the Heterochromatin Segments Revealed by CDAG-Banding and by FISH with 11 MSRs’ Probes in the Four Martes Species

Painting probes containing macrosatellite sequences revealed only some constitutive-heterochromatin sites in *Martes*. CDAG-staining clearly indicated that GC-enriched heterochromatin is situated in near-telomeric regions of short arms of several autosomes (Figure 1 and Figure 4, Figure 5 and Figure 6), where the studied MSRs were not localized. In addition, gaps in the fluorescent signal along a chromosome during the localization of the stone marten whole-chromosome–specific library (Figure 1 and Figure 4, Figure 5 and Figure 6) most likely pointed to the presence of other repeated sequences not covered by our study. A comparison of homologous syntenic groups among the four *Martes* species clearly revealed differences in morphology as well as individual patterns of distribution of CMA3-positive signals and of MSRs, for example, on homologs of MFO 5 and MFO 15 and MFO sex chromosomes (Figure 7). 

## 4. Discussion 

Now that significant progress has been made in sequencing and assembly of the coding part of the genome, approaches have begun to emerge that allow us to investigate various repeated DNA sequences. There is more and more evidence of the importance of repeated elements for the functioning of the genome. Nevertheless, this part of the hereditary material remains less studied as compared to nonrepetitive DNA. Here we used an array of modern molecular cytogenetic methods and bioinformatic genome analysis to describe distributions of constitutive heterochromatin and tandem repeats (telomeric sequences, macrosatellites, and ribosomal genes) on chromosomes of four species from the genus *Martes.*

Mustelidae (martens, minks, weasels, otters, badgers, wolverines, and others) is a diverse family that split off from other Carnivora ~17 million years ago [53]. Karyotypes of Mustelidae are a classic example of conserved genomes within the Canoidea clade, which were described at the dawn of the era of differential cytogenetics and have been confirmed to be conserved among several species by comparative chromosome painting [40,54]. The genus *Martes* (martens and sables) has a more basal position on the tree of Mustelidae. It has been hypothesized that the ancestor of all Mustelidae had a karyotype with diploid number 2n = 38, which is similar to chromosome sets of mustelids of the genus *Martes* (“*Martes*-like”), and this ancestral karyotype is still present in chromosome sets of sables and pine and stone martens [55]. Interest in the increased amount of constitutive heterochromatin in Mustelidae has encouraged research into satellite repetitive sequences’ content and distribution on their chromosomes in the 1980s [56]. This article presents for the first time data on tandem repeats in this family according to modern molecular cytogenetic and bioinformatic methods (Table 2 and Table 3). It is worthwhile to investigate karyotypes that are conserved in terms of the euchromatin fraction and show very similar G-banding patterns (Figure 1 and Figure 4, Figure 5 and Figure 6). In the comparative analysis of G-banding patterns in the karyotypes, some chromosomes look absolutely identical between the sable and the three studied marten species, while other chromosomes differ slightly. These differences are difficult to interpret: are they due to methodological factors, or do these differences indeed reflect existing diversity? Methods of molecular cytogenetics in combination with bioinformatic tools aimed at analyzing structural and functional features of each element of the chromosome set showed that even chromosomes having very similar G-banding patterns may contain different fractions of repeated DNA in different amounts (see, for example, homologs of MFO 1, 2, 3, and 4 in Figure 1 and Figure 4, Figure 5 and Figure 6).

The study of macrosatellites is relevant because they are involved in the organization of the genome and in the development of diseases [5]. In general, MSRs are poorly studied in animals, although macrosatellite D4Z4 is conserved, and homologous D4Z4 sequences have been identified in different animals [13]. This first attempt (our study) to identify MSRs in four representatives of the genus *Martes* suggests that the majority of chromosomes contain different repeat fractions in pericentromeric regions (Table 3). Given that only one individual of each species was subjected to this study, we cannot rule out the possibility of within-species variation of MSRs. It was found here that repeats S22, S30, S35, and S48 are present in regions homologous to MFO 5p and Y chromosomes in the species under study. Notably, these areas are chockful of different MSRs, each of which is also present in pericentromeric regions of some other autosomes. Satellite DNA is among the fastest evolving types of DNA [6], and this property can explain the dissimilar intensities of labeling of homologous regions among the different species when the same stone marten MSR FISH probe was employed.

We identified macrosatellite S40 only in an X-chromosome centromeric region. Apparently, this repeat may be at least *Martes*-specific because its probe did not label the X chromosome of the domestic cat. This finding contradicts the generally accepted view that the X chromosome is conserved among Carnivora taxa and retains the structure from a common ancestor [57,58]. Of note, of all the MSRs tested here, only the S46 probe manifested the noncentromeric interstitial localization on the q arm of the X chromosome aside from pericentromeric localization on the X chromosome and some autosomes.

The CDAG method for studying the distribution of constitutive heterochromatin in combination with MSRs’ localization paints a unique chromosomal “portrait” of each species in question owing to the diversity of locations, numbers, and sizes of clusters of repeated sequences (Figure 1 and Figure 4, Figure 5 and Figure 6). The sable, pine marten, and stone marten, which are united (on the basis of mitogenome analysis) into the subgenus *Martes*, share similar distributions of repeats, in contrast to the yellow-throated marten, which belongs to the subgenus Charronia and diverged ~5.8 million years before the present [59]. Of note, the sable and pine marten, which are considered sister species, have a distribution of satellite DNA sequences that is more similar to each other than in comparison with the stone marten.

In the studied genomes of *Martes,* there is a considerable number of various repeated sequences, and our work covers only a small proportion of them. Nevertheless, we show that even closely related species with a euchromatic part conserved between them have a specific set of core repeated sequences that contributes to variation of the amount and localization of constitutive heterochromatin. These varying repeats may contribute to many species-specific traits and adaptations to various ecological niches mastered by the species examined here. The interspecific diversity due to variation of the heterochromatin component has been described for pinniped karyotypes: a group of carnivores with a nearly unchanged ancestral karyotype [44].

Our results are consistent with the “library” hypothesis of satellite DNA evolution: closely related species share a set of conserved satellite DNA families that originated in an ancestor, but each of which is amplified differently among the species [60]. 

It was shown in the 1970s that heterochromatic blocks revealed by C-banding in animal genomes are composed of highly repetitive sequences (for example, [29,30,31,32]). Here we demonstrate that macrosatellites are the main component of the heterochromatic blocks in *Martes*. These most abundant macrosatellites likely form heterochromatic blocks in other animals too. The variations of types of macrosatellite sequences and their level of amplification are responsible for the species-specific variation of the size and distribution of heterochromatic blocks in the genome. The macrosatellites play an important formative role in chromosome structure by changing the size and shape of chromosomes and the genomic landscape. Ribosomal genes are also tandemly repeated and contribute to changes in chromosome structure. Here we demonstrate that ribosomal genes in *Martes* determine the variation of size and shape of the homologs (heteromorphism) of a NOR-bearing autosome pair (Figure 3 and Figure 7).

Current animal genome assemblies are not taking into account repetitive sequences. Therefore, the in silico–assembled genomes may benefit from comparative distribution maps of repetitive sequences, thereby resulting in a comprehensive picture of species chromosome organization by including information about heterochromatin clusters and their composition. Cytogenetic maps provide valuable information about chromosome size and about such varied-size structures as NORs, intercalary and pericentromeric heterochromatin, additional heterochromatic arms, and interstitial telomeric sites; these data are essential for the correct matching of an in silico genome and a physical chromosomal set.

## Figures and Tables

**Figure 1 genes-14-00489-f001:**
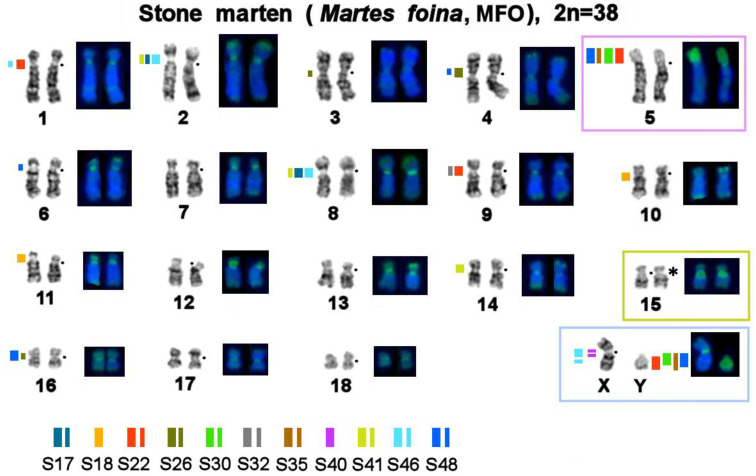
Localization of repetitive macrosatellite sequences in the genome of the stone marten. The metaphase plate was first stained with Giemsa after trypsin digestion for the G-banding and then restained with fluorescent dyes after formamide denaturation to reveal the composition of constitutive heterochromatin (CDAG-banding). A dot denotes a centromere position. * Clusters of ribosome genes (NOR). The thickness of a colored line reflects the relative intensity of a fluorescent signal. Autosomes 5 and 15 and sex chromosomes are marked with colored boxes for the comparison that is shown in Figure 7.

**Figure 2 genes-14-00489-f002:**
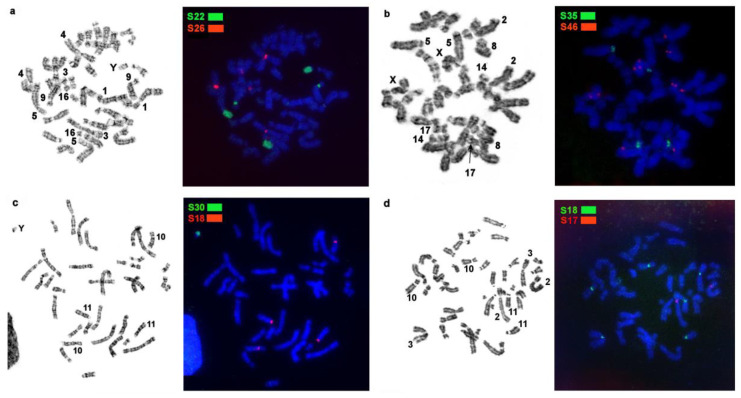
Examples of localization of stone marten repetitive macrosatellite sequences in the genomes of *Martes* representatives with GTG-banding of the same metaphase to the right: (**a**) macrosatellite probes S22 and S26 on stone marten (MFO) chromosomes; (**b**) macrosatellite probes S35 and S46 on pine marten (MMAR) chromosomes; (**c**) macrosatellite probes S30 and S18 on sable (MZIB) chromosomes; (**d**) macrosatellite probes S18 and S17 on yellow-throated marten (MFLA) chromosomes.

**Figure 3 genes-14-00489-f003:**
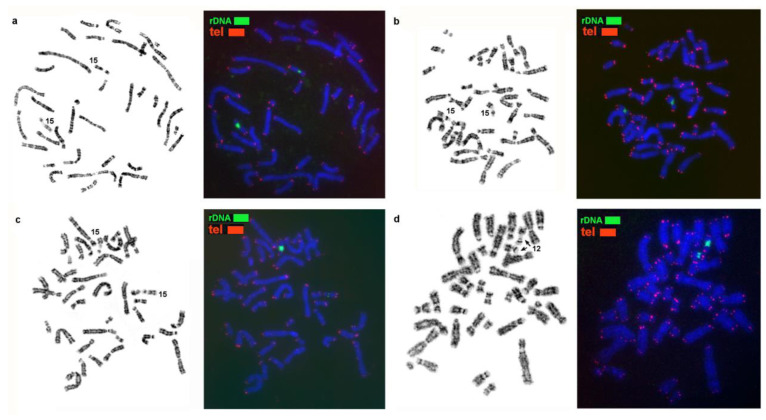
Localization of 5.8S + 18S + 28S ribosomal genes and telomere repeats on the chromosomes of (**a**) the sable (MZIB), (**b**) pine marten (MMAR), (**c**) stone marten (MFO), and (**d**) yellow-throated marten (MFLA). Arrows indicate NORs.

**Figure 4 genes-14-00489-f004:**
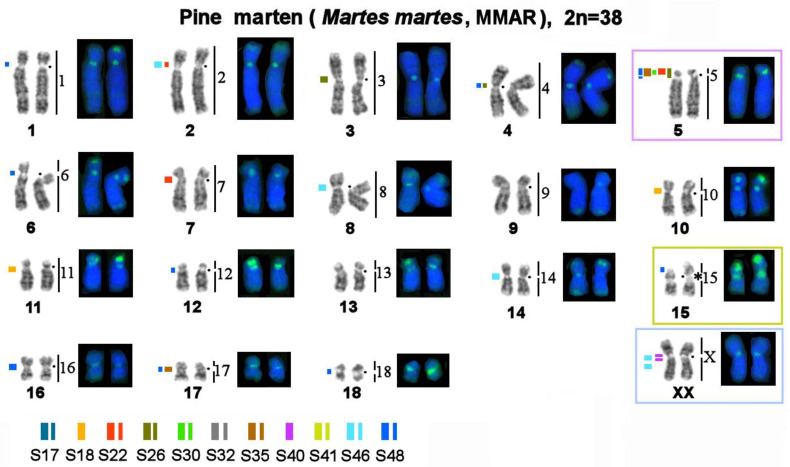
Distributions of repetitive macrosatellite sequences in the genome of the pine marten. The metaphase plate was first stained with Giemsa after trypsin digestion for the G-banding and then restained with fluorescent dyes after formamide denaturation to reveal the composition of constitutive heterochromatin (CDAG-banding). Numbers on the right indicate homology with stone marten (MFO) chromosomes. A dot denotes a centromere position. * Clusters of ribosomal genes (NOR). The thickness of a colored line reflects the relative intensity of the fluorescent signal. Chromosomes homologous to stone marten autosomes 5 and 15 and sex chromosomes are marked with colored frames for the comparison that is shown in Figure 7.

**Figure 5 genes-14-00489-f005:**
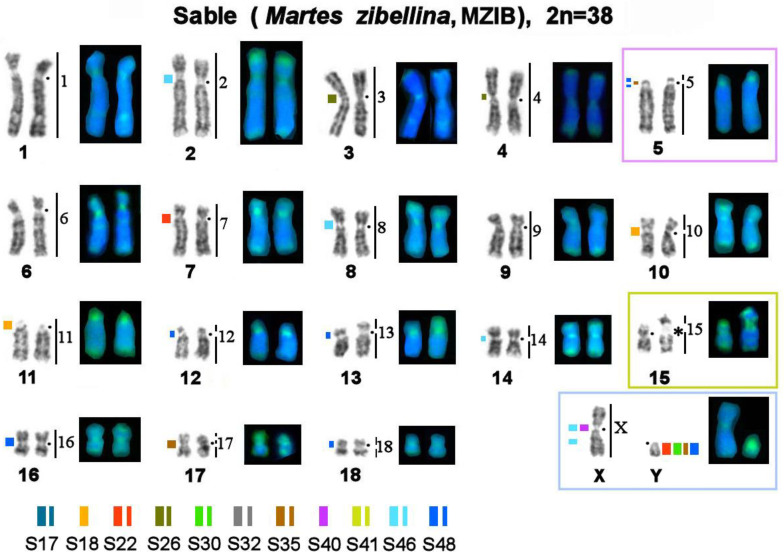
Localization of repetitive macrosatellite sequences in the sable genome. Symbols are the same as in Figure 4. * Clusters of ribosomal genes (NOR).

**Figure 6 genes-14-00489-f006:**
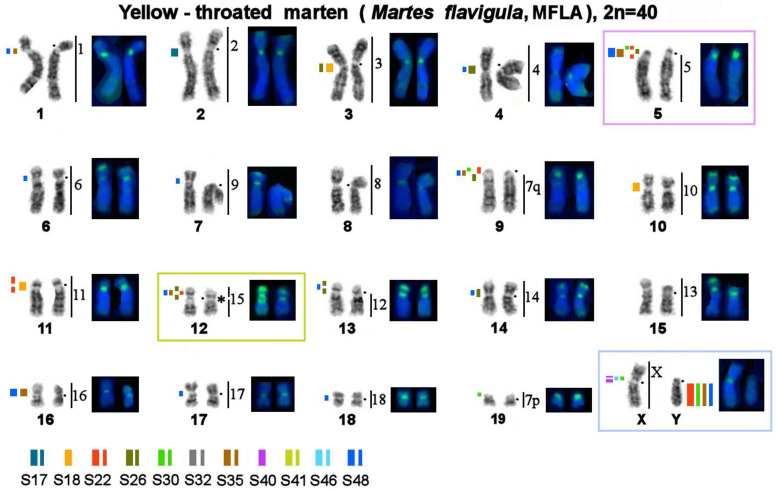
Localization of repetitive macrosatellite sequences in the genome of the yellow-throated marten. Symbols are the same as in Figure 4. * Clusters of ribosomal genes (NOR). On all Figures.

**Figure 7 genes-14-00489-f007:**
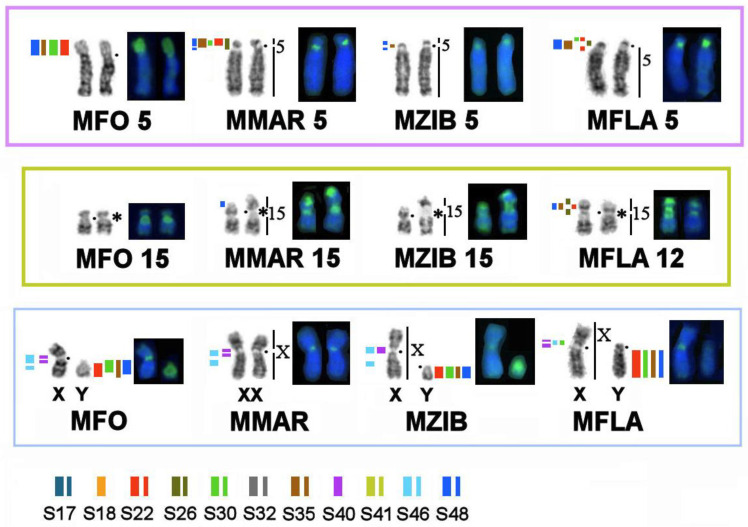
Differences in the distribution and composition of heterochromatin on homologous elements among the four *Martes* species as evidenced by CDAG-staining and FISH with macrosatellite probes. Symbols are the same as in Figure 4. * Clusters of ribosomal genes (NOR).

**Table 1 genes-14-00489-t001:** The list of the species analyzed in this study.

No.	Scientific Name	Code	2n	Sex	Common Name	Reference for FISH Data
1	*M. foina*	MFO	38	M	stone (beach) marten	[43]
2	*M. flavigula*	MFLA	40	M	yellow-throated marten	[39]
3	*M. martes*	MMAR	38	F	pine marten	this article
4	*M. zibellina*	MZIB	38	M	sable	this article

**Table 2 genes-14-00489-t002:** Most abundant satellite DNA sequences (used to develop the FISH probes) in the stone marten (*M. foina*) genome. N/A: not available. The letters “L” and “H” in repeat names indicate the prediction of TAREAN on the probability (L: low or H: high) of a tandem arrangement in the genome.

Repeat Name	Genome Proportion	Monomer Length, bp	GC Content	Homology to Known satDNA	Probe Length, bp
S17H	0.3%	714	60.22%	*Mustela vison* clone I225 microsatellite	410
S18H	0.28%	1041	70.61%	*Mustela putorius* 1080 bp Bam HI repeat	449
S22H	0.25%	986	69.68%	N/A	269
S26L	0.140%	1157	57.30%	N/A	425
S30H	0.12%	1580	68.35%	N/A	320
S32H	0.11%	1010	60.50%	N/A	440
S35H	0.11%	1148	68.12%	N/A	447
S40H	0.098%	2949	61.99%	N/A	400
S41H	0.096%	356	53.37%	*M. vison* clone I225 microsatellite	181
S46L	0.079%	1210	44.88%	N/A	267
S48H	0.073%	1146	65.27%	N/A	371

**Table 3 genes-14-00489-t003:** Localization of repeated sequences from the stone marten genome on chromosomes of the four *Martes* species: the stone marten, pine marten, sable, and yellow-throated marten. Chromosome ID numbers for syntenic groups in three species (MMAR, MZIB, and MFLA) are indicated in the stone marten chromosome nomenclature. ID numbers of chromosomes with weak signals are indicated in the regular font; boldfacing shows ID numbers of chromosomes having bright signals. Stone marten–specific repeats not detected by molecular cytogenetics in the three other species are highlighted in pink. Autosome-specific repeats are highlighted in green. An X chromosome-specific repeat—the macrosatellite localized only to the pericentromeric region of the X chromosome—is highlighted in blue.

Repeat Name	MFO, Male ♂	MMAR, Female ♀	MZIB, Male ♂	MFLA, Male ♂
S17H	2c, **8c**	n/a	n/a	**2c**
S18H	**10c**, **11c**	**10c**, **11c**	**10c**, **11c**	**3c**, **10c**, **11c**
S22H	**1c**, **5p**, **9c**, **Yq**	2c, **5p**, **7**	**7c**, **Yq**prox	5p, 5q, 7q-p, 11p, 11c, 15c, **Yq**
S26L	3c, **4c**, 6c	**3c**, 4c, 5p, 5c	**3c**, 4c	3c, **4c**, 5c, 7q-c, 12p, 12c, 14c, 15p, 15c
S30H	**5p**, **Yp**, **Yq**prox	5p	**Yq**prox	5p, 7qp, 7p-p, Xc, Yq
S32H	9c	n/a	n/a	n/a
S35H	5p, Ypq	**5p**, **7c**	5c, 17c, Yqprox	5c, 7q-c, 15p, 16c, Yq
S40H	**Xc**	**Xc**	**Xc**	**Xc**
S41H	2c, 8c, **14c**	n/a	n/a	n/a
S46L	1c, **2c**, **8c**, **Xc**, **Xq**int	**2c**, **8c**, **14c**, **Xc**, **Xq**int	**2c**, **8c**, 14c, **Xc**, **Xq**int	Xc
S48H	4c, **5p**, 6c, **16c**, **Yp**, **Yq**prox	1c, 4c, 5p, 5c, 6c, 12c, 15p **16c**, 17c, 18c	5p, 5c, 12c, 13c, **16c**, 18c, **Yq**prox	1c, 4c, **5c**, 6c, 7q-c, 9c, 12c, 14c, 15c, **16c**, 17c, 18c, **Yq**

## Data Availability

Raw sequencing data of *M. foina* genomic DNA were downloaded from NCBI SRA under accession number SRX8108270. Consensus sequences of *M. foina* 11 macrosatellites were submitted to the NCBI GenBank database under accession numbers OQ261720–OQ261730.

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
