# Peer review of "Maps of Constitutive-Heterochromatin Distribution for Four Martes Species (Mustelidae, Carnivora, Mammalia) Show the Formative Role of Macrosatellite Repeats in Interspecific Variation of Chromosome Structure"

_genes, 2023, doi:10.3390/genes14020489_

Round 1

Reviewer 1 Report

In the study “Constitutive heterochromatin distribution maps for four Martes species (Mustelidae, Carnivora, Mammalia) show the formative role of microsatellite repeats in the inter-specific variation of chromosome structure” the authors describe localization of bioinformatically identified microsatellite repeats (MSRs) for four closely related species of the Martes genus. MSRs were identified by FISH in stone marten (Martes foina); pine marten (Martes martes); yellow-throated marten (Martes flavigula) and sable (Martes zibellina). The study is methodologically sound and performed on rare samples, though is mostly descriptive.

Major points:

1.     Only one animal for each species is used for analysis. This aspect needs to be clearly indicated, and it is required discuss a possibility of within-species variation of MSRs localization.

2.     Abstract lacks significance statement.

3.     Section 3.2 lacks data supporting the statements – the only data shown is telomeric staining for yellow-throat marten (Fig. 2d). The supporting data needs to be presented  (possibly as Supplementary Information).

Minor points:

1.     The study would gain from more informative subheadings in the Results section, describing the findings of the section. (For example, not “Localization of MSR”, but “MSRs are localized to …”)

2.     Page 3, first paragraph:

“ Zoo-FISH confirmed the high conservatism of euchromatin

fraction in marten’s genomes in contrast to variations of

heterochromatin content”

That is an important statement for the study, but it lacks details and references. Which genomes were probed? Also specify what is Zoo-FISH.

3.     Page 3, first paragraph:

Comparison of human and domestic ferret

karyotypes revealed a low number of conserved segments in total – 32”

A low number of conserved segments in comparison to …? What is the “normal” number of conserved segments? Please clarify.

4.     Figures 1,3-5: Indicate how many metaphases were analysed. What are the coloured boxes?

5.     Table 1: Table legend lacks information on the letters H and L in repeat names.

6.     Table 3:

-       What is n/a?

-       Fill the empty cells for FCA. As there is only negative data for FCA, it might be good to take it completely from the table and leave only description in the text. However, the negative FISH data need to be presented (in Supplementary Information).

Author Response

Dear reviewer, thank you very much for your work in reviewing our manuscript! We tried to consider all the comments. After correction this manuscript was thoroughly edited by a professional language-editing service http://shevchuk-editing.com/index.html. We are sure that this improved the manuscript and made it more convenient for the perception of the data we received.

Reviewer 2 Report

Dear  Authors,

I have enjoyed reading your present communication, your study is well described and conclusions well sustained by the material and methods as well as the results presented. 

I do no have comments or observations for improve your MS.

Thanks,

Author Response

Thank you very much for the work with the manuscript and the positive feedback. We used a professional language-editing service http://shevchuk-editing.com/index.htm  to improve English language and style of our manuscript.

Round 2

Reviewer 1 Report

The manuscript had been significantly improved. A couple of points still need further attention:

1.     The point on within-species variation in the number of positions of repetitive elements.

Had there been any studies on within-species variation in Martes or other close species? Can the authors in the discussion speculate on the expected extend of this variation?  

The expert opinion here is important as it would define the limitations of the study and  clarify the future research possibilities for non-specialists.

2.     Images in Figure 7 are the fragments from Figs. 1, 4-6. It is convenient to show a few chromosomes of all four analysed species side-to-side, but to justify a duplication the new figure needs to carry some additional information; otherwise it is sufficient to refer to the previous figures.

3. Add supporting data to the statement in section 3.4:

“In addition, gaps in the fluorescent signal along a chromosome during the localization of the stone marten whole -chromosome-specific library most likely pointed to the presence of other repeated sequences not covered by our study. “
